# Trait Energy and Fatigue Influence Inter-Individual Mood and Neurocognitive Responses during Work Done While Sitting, Standing, and Intermittent Walking: A Randomized-Controlled Crossover Design

Hannah M. Gigliotti [1], Cody Hodgson [2], Mary Riley [2], Brittany Marshall [2], Christie L. Ward-Ritacco [3], Joel Martin [4] and Ali Boolani [5],*

[1] Department of Biology, Clarkson University, Potsdam, NY 13669, USA
[2] Department of Physician Assistant, Clarkson University, Potsdam, NY 13669, USA
[3] Department of Kinesiology, University of Rhode Island, Providence, RI 02881, USA
[4] Sports Medicine Assessment Research & Testing (SMART) Laboratory, George Mason University, Manassas, VA 22030, USA
[5] Honors Department, Clarkson University, Potsdam, NY 13669, USA
* Correspondence: aboolani@clarkson.edu

**Abstract:** College students can be sedentary for a majority of the day, which may exacerbate mental health issues or lead to declines in cognitive task performance; however, interventions to address sedentary behaviors may not positively influence everyone. Therefore, the present study sought to identify inter-individual cognitive performance and mood changes of college students during the performance of a cognitive task battery, while seated, standing and with intermittent bouts of walking. Participants (n = 31, age = 25.80 ± 3.61 yrs, 7 male) completed a series of baseline questionnaires including the Trait Mental and Physical Energy and Fatigue survey. Using a randomized controlled cross-over design, participants completed 3 separate testing sessions. At each session, they performed a series of three rounds of cognitive tasks for 27 min and self-reporting mood states for 1 min in the seated position. Each round of cognitive testing was followed by a 2 min break. Each testing day had participants spend the 2 min break in a different condition: sitting, standing, or walking. A series of mixed ANOVAs were used for the primary analysis and a combination of machine learning regressors and classifiers were used for the secondary analysis. Our results suggest that there are unique inter-individual responses to each of the interventions used during the 2 min break. Participants who were low-trait mental and low-trait physical energy benefited the most from the standing desk intervention, while also reporting significant benefits of intermittent walking. However, participants who were low-trait mental fatigue had significant negative consequences of using both standing desks and walking intermittently, while those who were high-trait mental fatigue saw no change in cognitive responses or moods in those conditions. Post hoc machine learning analyses had modest accuracy rates (MAEs < 0.7 for regressors and accuracy rates >60% for classifiers), suggesting that trait mental and physical energy and fatigue may predict inter-individual responses to these interventions. Incorporating standing desks into college classroom settings may result in some students receiving cognitive benefits when inter-individual variability in mood and cognitive responses are accounted for.

**Keywords:** anxiety; standing desks; interindividual differences; fatigue; college students; sedentary behavior; moods

## 1. Introduction

Significant evidence demonstrates that young adults, especially during their college years, may experience significant mental health challenges [1,2]. Prior to the COVID-19 pandemic, the prevalence of mental disorders among college students was estimated

to be 25% [3] and the pandemic seems to have exacerbated mental health issues in this cohort [4,5]. While pre-pandemic evidence existed that college students were more likely to suffer from mental health issues than their age matched peers [6], the pandemic has almost doubled the number of undergraduate and graduate students who report feeling anxious and/or depressed [7]. While university administration and support staff can develop programs and approaches to help students cope with their individual identities and reduce institutional stressors through the creation of more inclusive environments [8], these large-scale changes require time and resources for implementation. More immediate change in student mental health status may occur if individual faculty members are able to enhance the classroom environment during instructional time.

A feasible method includes reducing sitting time during classroom instructions, as significant evidence exists showing that performing cognitively demanding tasks while seated significantly increases feelings of anxiety, depression, and reduces cognitive task performance in young adults, even when that sitting time is limited to 1.5 h [9–11]. Interestingly, in a study by Boolani and colleagues [11], it was reported that there was a significant increase in feelings of anxiety, depression, fatigue and anger after the performance of only 22 min of cognitively demanding tasks. These findings suggest that college students who sometimes spend 50 min to 3 h in a classroom setting performing cognitively demanding tasks while seated may experience an acute negative change in moods and declines in cognitive performance. Individual college students often take steps to stay alert during class or while studying, such as consuming caffeine [12,13]. However, evidence exists that caffeine consumption may exacerbate feelings of anxiety in college students who perform cognitive tasks in a seated position [10,11]. Therefore, other behavior-based options should be explored.

The implementation of standing desks in classroom spaces may be one such strategy. Evidence from non-college students suggests that the use of standing desks may improve physiological outcomes, including postprandial glucose, HDL cholesterol, and anthropometrics, although at present, the summary of the evidence to support the use of standing desks to improve mood outcomes has provided mixed results [14]. Another easy to implement solution for faculty members and students to improve mood and cognitive performance may be to implement movement breaks during class time. Breaking up sedentary time into smaller time periods by encouraging students to be physically active, for example, by walking intermittently for a set time frame, may reduce the impact of continuous cognitive demands on moods and cognitive performance. Evidence from school age students suggests that performing physical activity during class time improves health, learning outcomes, and various aspects of cognition [15]. This may be an effective strategy for improving college student health; however, we are unaware of any studies examining the impact of breaking up sitting behavior with physical activity in this cohort.

It has been shown that a single bout of physical activity can impact both mood [16–18] and cognition [19] positively; however, the duration and type of physical activity both influence the magnitude of the effect. Recently, a study by Carmichael and colleagues [20], reported that four minutes of stair walking did not modify feelings of energy and fatigue. Unfortunately, this study did not measure any other mood states. Interestingly, in a study of older adults, Boolani and colleagues [21] found that 6 min of self-paced walking attenuated the negative effects of performing 2.5 h of sustained cognitive task performance while sitting. Participants self-reported significantly improved moods compared to both before and after the performance of cognitive tasks [21]. These findings suggest that self-paced walking may attenuate the negative impact of sustained cognitive task performance on moods. To our knowledge the current literature is lacking studies that have used short bouts of walking to assess if these could attenuate the negative impact of performing multiple short bouts of cognitive tasks.

Despite significant evidence from multiple meta-analyses on the positive impact of a single bout of physical activity on moods [16–18], recent evidence suggests that not everyone benefits from the mood-enhancing benefits of a single bout of exercise [22,23]. Recent work

suggests that trait (long-standing pre-disposition) level mental and physical energy and fatigue may explain the inter-individual responses in moods and cognitive task performance after the consumption of caffeine [24] and an adaptogenic-rich caffeinated beverage [25]. Further, cross-sectional data suggest that trait level energy and fatigue explain associations between moods and various health-related behaviors [26–28]. Although we are unaware of why trait mental and physical energy and fatigue modify responses to various interventions, evidence exists that prolonged states of energy and fatigue have distinct yet overlapping epigenetic markers [29], and trait mental and physical energy and fatigue are associated with unique gut microbiome [30]. Further, studies that have examined state level energy and fatigue suggest that dopamine [31], Annexin A1 [32], and peripheral mitochondrial function [33] are associated with changes in feelings of energy, while histamine [34], serotonin [31], and TNF-$\alpha$ [31] are associated with feelings of fatigue. In addition to biological differences, evidence also exists that both trait and state level feelings of energy and fatigue are associated with different aspects of gait and balance, suggesting that these moods may by associated with different aspects of the frontal cortex [35–39]. Recently, Filippi and colleagues [40] provided evidence on the importance of measuring trait, prolonged state, and state level energy and fatigue, as trait may uniquely influence prolonged state and state level energy and fatigue. Based on this evidence, and the fact that the survey to measure trait level mental and physical energy and fatigue is a short 30 s survey, the current study chose to use trait mental and physical energy and fatigue to identify inter-individual responses to the selected interventions.

Considering the existing literature, the objective of this study was to identify inter-individual responses in cognitive task performance and moods during completion of 1.5 h of cognitive tasks performed while in a seated position, a standing position, and while walking intermittently within a group of college students. This study also had a secondary post hoc exploratory objective of trying to use trait mental and physical energy and fatigue to identify changes in cognitive task performance and moods for the three interventions using machine learning.

## 2. Methods

### 2.1. Study Design

A randomized—controlled, within–in participants, crossover design was used to examine the effects of three treatment conditions, sitting, standing, and intermittent walking, on neurocognitive performance. In all conditions, the mental task battery was administered at three time points (3–30 min, 34–61 min, 65–92 min). Specifically, in the sitting condition, participants completed 28 min of a mental-task battery (described below), followed by 2 min of seated rest, prior to engaging in the next bout of mental tasks, and repeated this for 3 total mental task battery trials. In the standing condition, participants completed the same 28 min mental-task battery followed by 2 min of standing rest, prior to engaging in the next bout of mental tasks, and repeated this for three total mental task battery trials. Finally, the intermittent walking condition consisted of individuals performing the mental task battery for 28 min followed by 2 min of self-paced walking in the hallway outside of the lab prior to engaging in the next bout of mental tasks and repeated this for three total mental task battery trials. Participants completed each condition in a randomized order, after the completion of a familiarization session. On average, participants began the study 3.0 ± 2.1 days after their familiarization visit.

### 2.2. Screening

Participants were recruited using campus-wide emails and verbal announcements in large classes (>20 students) at a small private university in Northern New York, and through the posting of flyers with QR codes throughout campus, directing students to complete the online inclusion/exclusion criteria survey (SurveyMonkey, Inc., San Mateo, CA, USA, www.surveymonkey.com). To be included in this study, participants had to be between 18 to 45 years of age; have the ability to ambulate unassisted for two minutes; be

able to stand unassisted for 90 min; and be a current undergraduate or graduate student. Exclusion criteria included presence of chronic medical conditions (including heart disease, diabetes); medication use for chronic condition(s) with the exception of oral contraceptives; acute illness at the time of testing; neurological conditions; lower-extremity injury within the last two weeks that may impair the participant from standing for a prolonged period; use of any supplements with the exception of vitamins and minerals; and color blindness.

### 2.3. Participants

Approval for the study was granted by the Clarkson University, Institutional Review Board (approval # 18–49.2). Volunteers not excluded by the screening were invited to the testing facility. All participants read and signed the approved consent form. Participants were informed that they would be participating in a study investigating the effects of sitting, standing desks, and intermittent walking on mental performance and moods.

Of the 40 volunteers who responded to the recruitment, six were excluded due to not meeting inclusion/exclusion criteria (1 for having color blindness, 2 for having chronic neurological conditions, 1 for an acute respiratory infection, and 2 for not being able to stand for 1.5 h). Of the 34 who qualified for the study, 31 participants completed the study (7 males, 24 females, age = 25.80 ± 3.61 years). Using G*Power (version 3.1.9.6, Heinrich-Heine-Universität Dusseldorf, Germany), an a priori power analysis was completed and showed that a sample size of 27 would provide statistical power ($\alpha = 0.05$, 1-$\beta = 0.80$) to detect a 2-group x 3-intervention x 3-time interaction effect size of 0.35, assuming a correlation across the repeated measures on time of 0.50. To reduce the potential for Type II errors, 31 participants completed the study in case there were outliers and data had to be excluded. Accordingly, no data were excluded from this study. Participant characteristics can be found in Table 1.

**Table 1.** Participant characteristics (n = 31).

| Measure | Mean (SD) |
|---|---|
| Sex (male:female) | 7:24 |
| Height (cm) | 172.44 (9.08) |
| Weight (kg) | 74.87 (14.5) |
| Age (years) | 25.8 (3.6) |
| Trait Physical Energy | 6.06 (1.82) |
| Trait Physical Fatigue | 5.06 (2.0) |
| Trait Mental Energy | 5.26 (1.76) |
| Trait Mental Fatigue | 5.81 (2.14) |
| Vigorous Physical Activity (h/week) | 2.40 (5.27) |
| Moderate Physical Activity (h/week) | 4.17 (10.02) |
| Sitting Time (h/day) | 9.47 (4.00) |
| Pittsburgh Sleep Quality Inventory Score | 6.94 (2.1) |
| Caffeine Consumption (servings per day) | 1.8 (2.2) |

The average self-reported nightly sleep during the month prior to the study was 7.52 ± 1.80 h and the number of hours of self-reported sleep the night before each testing session did not significantly differ between conditions ($p = 0.807$); sitting (7.44 ± 1.81 h), standing (7.61 ± 1.38 h), intermittent walking (7.71 ± 1.62 h). Of the 31 participants, 20 reported being physically inactive over the last 7 days prior to the start of the study (0 min of vigorous or moderate physical activity), 18 of the 24 women in this study self-reported the use of birth control, and 23 of the 31 participants had Pittsburgh Sleep Quality

Inventory (PSQI) scores > 5. All participants reported being consumers of caffeine; however, all participants self-reported abstaining from caffeine use for at least 12 h prior to testing.

### 2.4. Baseline Measures

Trait mental and physical energy and fatigue: Using the second section of the State and Trait Mental and Physical Energy and Fatigue survey [41], the trait aspect (long-standing pre-disposition) of mental and physical energy and fatigue was collected during a familiarization visit. The trait component references how the respondent usually feels and contains a 12-item scale with 3 items for each of the 4 trait outcomes. Representative statements include "I feel I am full of pep" and "I have feelings of being worn out." Participants are asked to mark how they normally feel on a 5-point scale ranging from "never" to "always." In previous studies, Cronbach's alpha coefficients range from 0.73 to 0.94 [41,42]. With the current data, the alpha coefficients ranged from 0.79 to 0.86 (Trait Physical Energy = 0.81, Trait Physical Fatigue = 0.79, Trait Mental Energy = 0.86, Trait Mental Fatigue = 0.85).

Sleep quality and quantity: The Pittsburgh Sleep Quality Inventory (PSQI) was used to measure self-reported sleep quality and quantity over the last 30 days. The 19-question survey assesses seven dimensions of sleep: sleep quality, latency, duration, habitual sleep efficiency, sleep disturbance, use of sleep medication, and daytime dysfunction. The scores for each dimension range from 0 to 3 and are summed to create an overall sleep quality score, with higher scores indicating worse sleep quality. Previous studies have defined poor sleep quality as PSQI scores > 5 [43]. For the current study, we used data from the sleep duration component of the PSQI to identify the participants' self-reported average night sleep to determine if participants had received adequate sleep the night before the study.

Physical Activity: The International Physical Activity Questionnaire—Short Form (IPAQ-SF) was administered to assess self-reported physical activity levels and sitting behavior [44]. Respondents were asked to report how many days they perform vigorous (i.e., heavy lifting, digging, aerobics, or fast bicycling), moderate (i.e., carrying light loads, bicycling at a regular pace, or doubles tennis) and light (i.e., walking at work or at home, walking to travel from place to place) physical activity and the number of hours and minutes per day engaged in each of these types of activities. Physical activity levels are calculated by multiplying the number of days per week of physical activity performed by the number of minutes. Results for this study are presented in hours per week. Respondents are also asked to identify the days per week, in addition to the number of hours and minutes per day spent sitting (including time spent sitting at a desk, visiting friends, reading or sitting or lying down to watch television) on a typical weekday and weekend day. Sitting time for this study was calculated by multiplying the number of self-reported weekdays spent sitting by the time spent sitting on weekdays, multiplying the number of hours and minutes spent sitting on a typical weekend day by two, then adding the weekday and weekend results for total sitting time. Results are presented as number of hours per day of sitting time.

Caffeine consumption: Respondents completed a questionnaire that asked them to identify their consumption of different caffeinated beverages and edible products containing caffeine. Consumption of caffeinated products was standardized to 8oz servings [42] and the total amount of caffeine in servings per day was calculated for each participant.

### 2.5. Testing Day Measures

Pre-testing measures: Prior to the start of testing, participants were asked to complete a series of surveys. The sleep survey asked about prior night's sleep, including the time they went to sleep, the time they woke up, the number of times they woke up in the middle of the night, and for how long they were awake if they woke up in the middle of the night. The data were then used to calculate self-reported prior night's sleep [10,11]. Participants were also asked to identify the amount of moderate and vigorous physical activity they had performed over the last 12 h and were asked to self-report food and beverage intake

over the past 24 h. Participants who had performed moderate or vigorous physical activity or had consumed caffeine over the last 12 h did not complete testing that day and were re-scheduled.

Mental Task Battery: Participants completed the following mental task battery using a 12.9-inch Apple iPad Pro (256GB, model number ML0T2LL/A) with an attached New Trent iPad Pro QWERTY keyboard (model number NT88B). In all conditions, the iPad Pro was placed at a 15-degree angle on a Power portable Laptop Table Desk Adjustable Riser (PWR+, model number 941-PWR50-754520) stand-up desk that could be adjusted in both a seated or standing position. All tasks were performed using the Membrain Platform (PsychTechSolutions, Potsdam, NY, USA). The order of tasks can be found in Figure 1.

(1) Serial Subtraction 3 and 7 task—Participants were asked to silently subtract backwards in threes or sevens from a random number starting between 800 and 999 that was presented on the iPad screen (Tahoma Regular fond; size 20). Participants were instructed to type their answers as quickly and as accurately as possible. After each answer was submitted by the participant, the answer was cleared from the screen and participants continued to subtract three or seven from their previous answer. Participants were allowed to complete as many attempts as possible in two minutes. In the case of incorrect responses, subsequent responses were scored as correct if they were correct in relation to the new number. The tasks were scored for the number of correct responses, the number of total attempts, and the percentage of correct responses (correct responses/total responses X 100) [11,45,46].

(2) Continuous Performance Task (CPT)—Participants monitored a continuous series of letters (A–Z; Tahoma Regular font, size 20 point) presented on the screen for 1000 ms and were asked to respond to the detection of the letter "X" only when preceded by the letter "A" by tapping on the screen or pressing the "Space" key. The participants used the same modality for all tests over the course of the study. A total of 48 correct targets were randomly presented over the course of two minutes, and the task was scored for the percentage of target strings correctly detected, errors of omission (missed targets), false alarms (key pressed/screen tapped when no target presented), and the average reaction time for correct decisions [10,11,45,46].

(3) Rapid-Visual Input Processing Task (RVIP)—Participants were required to monitor a continuous series of digits (1–9; Tahoma Regular font, size 20 point) presented on the screen every 1000 ms. The participants were given a primary, secondary, and tertiary task. The participant's primary task was to detect the presentation of three consecutive odd digits that were in ascending order (e.g., 3–7–9), the secondary task was to detect three consecutive even digits that were in ascending order (e.g., 2–4–8), and the tertiary task was to identify the number 6 when it was presented on the screen. The participants were given the option of using the keyboard or the touch screen on the iPad and participants used the same modality over the course of the study. If participants chose to use the touch screen, they were asked to press the green button for the primary task, the red button for the secondary task, and the yellow button for the tertiary task. If participants chose to use the keyboard, they were asked to press the left arrow for the primary, the right arrow for the secondary, and the up arrow for the tertiary task. Of the 480 stimuli presented, there were 8 primary targets, 8 secondary targets, and 48 tertiary targets over the course of the 8 min protocol. The task was scored for the number of correct detections of each target, average reaction time for correct detection of each target, the number of false alarms for each task, and errors of omission (missed targets) [10,11,45,46].

(4) Concentration Task Grid (CGT)—A 100 square grid was used as a measure of concentration. The grid is arranged in a $10 \times 10$ square with each square containing a two-digit number (from 00 to 99) which is randomly placed in the center of each of the squares. Participants were asked to mark off as many consecutive numbers as possible starting from 00 within a one-minute period, by tapping on the desired square [47].

(5) Distractors—Participants were presented target arrows, in either red or green, every 1000 ms. Prior to the start of each test, participants were informed whether the color of the target for the test was red or green. Participants were asked to press the direction of the arrow presented on the screen only if it was a target. For example, If the target was red and the arrow presented on the screen was red, then participants pressed the arrow on the keyboard that corresponded with the direction of the arrow presented on the screen [48]. A total of 12 distractors were presented during each one-minute test and participants were scored for correct responses, incorrect responses, and average response time for each correct response.

(6) Paced Visual Serial Addition Task (PVSAT)—Participants were asked to start at the number 0 and add the digit shown on the screen (0–9). Participants were then asked to remember the previous number and add the next digit presented on the screen and type their responses on the keypad [49]. After each entry, a new digit was presented. Participants were given one minute to add as many digits as possible. In the case of incorrect responses, subsequent responses were scored as correct if they were correct in relation to the new number. Participants were scored for the number of correct responses and the average time it took to complete each correct response.

(7) Stroop Task—Participants were presented words (Tahoma font, 20 point), every 1000 ms, on a screen spelling the names of colors (i.e., blue, yellow, red, green), which were either congruent (words presented in the same color as the color they were spelling out, such as the RED presented in red font) or incongruent (words presented in a color different from the color that they spelled out, such as the word RED spelled in green font). Participants were presented with all five options of colors used in this test (red, green, yellow, black, and blue) and asked to press the color of the text (font color) and not the word presented on the screen. There were 30 congruent and 30 incongruent tasks presented in random order and participants were scored on the number of correct responses, number of incorrect responses, and the average reaction time for correct responses. Each test was scored separately for both congruent and incongruent tasks [50,51].

(8) Eriksen Flanker Task—During this task, participants were first presented with a white fixation cross for 200 ms, followed immediately by five equally sized arrows arranged in a 10.5 cm horizontal array for 800 ms. Participants were instructed to attend to the central arrow and ignore the four flankers. Participants were asked to press the left key if the central arrow was facing left and the right arrow if the central arrow was pointed to the right. If all arrows were pointing the same direction (e.g., "< < < < <"), then the trial was considered congruent. However, if the central arrow was pointing in the opposite direction (e.g., "> > < > >") then the trial was considered incongruent. Subjects were presented with 30 congruent and 30 incongruent trials in random order. The tests were scored on the number of correct responses, number of incorrect responses, and the average reaction time for correct responses. Each test was scored separately for both congruent and incongruent tasks [52].

(9) Perceptual (Fast Count) Task—Participants were asked to press the number key that qualified with the number of dots presented on the screen. Each dot was approximately 0.5 cm in diameter and the number of dots on the screen ranged from 4 to 7. The number of dots was randomized across each trial and trials lasted 1 min. Participants were scored based on number of correct responses, average reaction time for each correct response and the total number of trials conducted in one minute [53].

(10) Modified Corsi-Block and Visuo-Spatial Memory Task—A pilot test using a combination of the Corsi-Block test and a Visuo-Spatial Memory task was designed to test participants' visuo-spatial memory. Participants were presented a 3 × 3 square grid on the screen, where a dot was flashed in each of the grids in a randomized order and in a randomized location (i.e., middle of the top right square in the grid, bottom left corner of the top right square in the grid). The dot remained on the screen for 500 ms before the next dot appeared. After the entire grid had dots present and the last dot

had remained on the screen for 500 ms, the dots were removed, and the participants were asked to identify the order in which the dots appeared on the screen and the location where they appeared. Each participant completed 10 trials. Due to poor test–retest reliability, this test was not included in the results of this study.

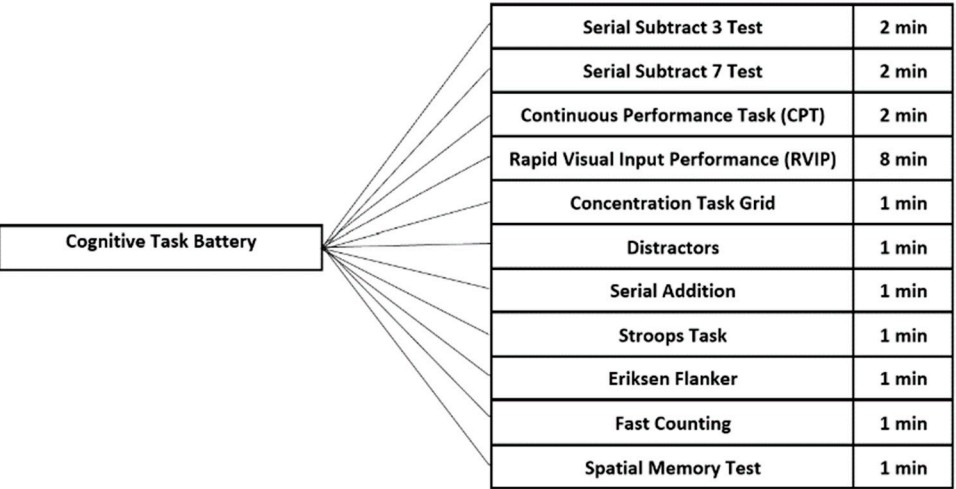

| Cognitive Task Battery | Serial Subtract 3 Test | 2 min |
| --- | --- | --- |
| | Serial Subtract 7 Test | 2 min |
| | Continuous Performance Task (CPT) | 2 min |
| | Rapid Visual Input Performance (RVIP) | 8 min |
| | Concentration Task Grid | 1 min |
| | Distractors | 1 min |
| | Serial Addition | 1 min |
| | Stroops Task | 1 min |
| | Eriksen Flanker | 1 min |
| | Fast Counting | 1 min |
| | Spatial Memory Test | 1 min |

**Figure 1.** Order of Tasks During Cognitive Task Performance.

*2.6. Mood and Motivation Surveys*

(1) Motivation survey: A 100-point Visual Analog Scale (VAS) was used to assess the participant's motivation to perform mental tasks. The scale was anchored with "No motivation at all" and "Highest motivation ever" [10,11].

(2) Profile of Mood Survey Short Form (POMS-SF): The 30-item Profile of Mood Survey Short Form was used to assess current mood states on a five-point scale ranging from "Not at all" (scored as 0) to "Extremely (scored as 4). Depression, tension/anxiety, anger, fatigue, and vigor (energy) were scored as a sum of five variables (i.e., vigor (energy) = energetic + lively + full of pep + vigorous + active) with scores ranging from 0 to 20. Confusion was the sum of four variables and then, the variable efficient was subtracted from it (i.e., confusion = confused + muddled + bewildered + forgetful – efficient), with scores ranging from −4 to 16. A total mood disturbance score was also calculated by adding depression, tension, anger, fatigue, and confusion and subtracting vigor (energy) from the total score. Among healthy adults, the Cronbach's alpha is reported to be 0.90 [54]. For the current study, the Cronbach's alpha ranged from 0.69 to 0.81 (Tension/Anxiety = 0.781, Depression = 0.766, Anger = 0.777, Fatigued = 0.808, Vigor = 0.767, Confused = 0.690).

(3) State Mental and Physical Energy and Fatigue: Using the third section of the State and Trait Mental and Physical Energy and Fatigue survey [41], current state moods of mental and physical energy and fatigue were assessed using a 12-item VAS, with each state outcome containing three items. Each item was anchored by "No feeling at all" to "The highest imaginable feeling". The Cronbach's alpha for these moods range from 0.88 to 0.90 [24–26,30,42]. For the current study, the Cronbach's alpha ranged from 0.91 to 0.96 (state physical energy = 0.940, state physical fatigue = 0.909, state mental energy = 0.959, state mental fatigue = 0.943).

*2.7. Procedure*

After participants were screened, they were invited to the lab to complete the familiarization day. At the beginning of the familiarization day, all participants completed a written informed consent prior to engaging in any study procedures. Following the completion of the informed written consent, participants' height was measured using a stadiometer and weight using a digital scale (Tanita TBF-410, Tanita Corporation, Tokyo,

Japan). To reduce the risk for learning effects during the mental task battery, participants completed the pre-testing questionnaire, one mental task battery and one set of mood and motivation surveys. Participants were then provided instructions regarding their testing days, including directions to abstain from caffeine and vigorous and moderate physical activity for a minimum of 12 h and to get their usual night's amount of sleep.

Participants were then scheduled for 3 testing sessions, with each session being a minimum of 48 h apart, but within 14 days of the previous session. The average number of days between sessions was $6.0 \pm 2.68$ days. To control for diurnal variations, participants were scheduled $\pm 30$ min from the time of their first testing day (i.e., if the first testing day was scheduled at 1100, then the subsequent two testing days could be scheduled between 1030 and 1200) [55]. Since sleep loss has substantial effects on mood and cognition [55], participants who reported 2 h more or less than their usual sleep duration (taken from the PSQI) were not tested that day and re-scheduled. Participants reporting consuming caffeine and/or participating in moderate or vigorous physical activity 12 h prior to testing were also re-scheduled.

On testing days, participants came into the lab where they completed the pre-testing questionnaire to determine testing eligibility prior to beginning the testing day. Participants were randomly assigned to a condition, and they were asked to complete a mental performance battery followed by mood surveys, and a 2 min rest period. This was repeated 3 times and the entire duration of the testing day was ~92 min (Figure 2). During administration of the mood survey, facial feature data were collected using the X-Box Kinect V2 RGB and Depth sensor camera (Microsoft Corporation, Seattle, WA, USA, model number XOne KinectSensor CR).

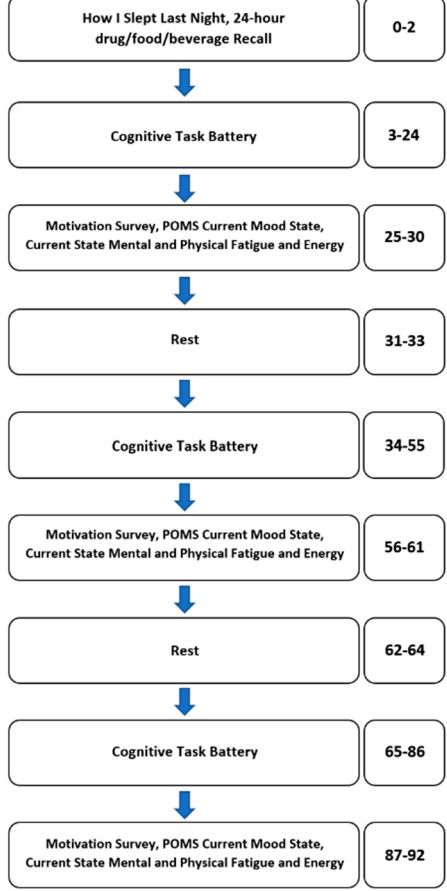

**Figure 2.** Task and Time. Note: Completion times of each task are means of the participants. Rest times were a standard 2 min for all participants.

*2.8. Statistical Analyses*

Pre-processing: Cognitive data from the Membrain platform and survey data from SurveyMonkey were downloaded into Excel, scored, and exported into Python (version 3.9, Python Software Foundation, Wilmington, DE, USA) where data were merged. The author who performed the initial analyses (H.G) was blinded to the conditions and all preliminary statistical analyses were performed prior to breaking the blind. Using a combination of scatterplots, descriptive statistics, and the Wilks–Shapiro test ($p > 0.05$), normality was assessed. Normalization techniques were applied to transform variables that were not normally distributed. Using the pingouin library [56], an initial 3 (day) $\times$ 3 (time) repeated measures Analysis of Variance (RM-ANOVA) to understand if there were any learning effects. No significant differences were noted for any measures ($p > 0.05$) between days. The median for each trait variable (Trait Physical Energy = 6, Trait Physical Fatigue = 5, Trait Mental Energy = 6 and Trait Mental Fatigue = 6) was identified and then surrogate variables were created to represent these in dichotomous format with values of 1 assigned to the 50th percentile and 2 otherwise.

Primary Analysis: Using the rpy2 library [57] in Python, the afex package (version 1.2) was accessed in R [58] to examine significant interactions between 2 levels of traits (low v. high) $\times$ 3 interventions (sitting, standing, intermittent walking) $\times$ 3 time points (3–30 min, 34–61 min, 65–92 min) using a mixed ANOVA. No significant outliers were detected and variables that exhibited a poor fit were not removed or transformed. The justification for this comes from the nature of the repeated measures format where transformation of one measure must be evenly applied to all other measures of that variable, which would exacerbate analytical issues and make interpretation more difficult. Further, we are unaware of any non-parametric test that is equivalent to a $2 \times 3 \times 3$ mixed ANOVA. Violations of sphericity were accounted for using the Greenhouse–Geisser correction to degrees of freedom [59]. If any of the results were significant, post hoc analyses were conducted using a combination of $2 \times 2$ repeated measures ANOVAs, one-way ANOVAs, between- and within-subjects t-tests, Kruskal–Wallis, Mann–Whitney U and estimated means. All post hoc results were adjusted using a Benjamini–Hochberg False Discovery Rate (FDR) of 20% [60]. Due to the study design, generalized effect sizes were calculated for each result and are presented [61,62]. Additionally, due to the large number of primary analyses, a Benjamini–Hochberg FDR was applied to all primary analyses to correct for Type I errors and a 20% FDR was considered acceptable [60]. If missing data were encountered for a test, the participant was removed from the analysis. All $\alpha$ presented values are adjusted. An adjusted $\alpha$ of 0.05 was used to justify significance.

Post hoc machine learning analyses: To complete the exploratory machine learning analyses, the sklearn library was used. A Principal Component Analysis (PCA) was performed on the 4 trait measures. The components were set to explain 90% of variance. Using the Kaiser–Gutmann criterion, which recognizes PCs with an Eigenvalue $\lambda > 1.0$ as significant [63]. Data were then imputed for missing data using the Multiple Imputation by Chained Equation (MICE) Method [64]. Due to the small sample size, data were split to compare interventions (sitting vs. standing, standing vs. walking, and sitting vs. walking). All data were scaled and a combination of regressor and classifier models were used. Regressor models, Linear Regression, Decision Tree Regressor, K Neighbors Regressor, AdaBoost Regressor, Linear Support Vector Regressor, Gradient Boosting Regressor, Random Forest Regressor, Kernel Ridge, Stochastic Gradient Descent Regressor, Elastic Net Regression, and LASSO, were used to determine whether each trait alone, a combination of two traits, the PCAs of the traits and the interventions alone could accurately predict the total change from time point 1 to time point 3 for the various outcomes as well as the PCAs of the changes in outcomes. Classifier models, Random Forest Classifier, K Neighbors Classifier, Support Vector Classifier, Nu Support Vector Classifier, Decision Tree Classifier, Random Forest Classifier, Gradient Boosting Classifier, AdaBoost Classifier, Guassian Naïve Bayes, Linear Discriminant Analysis, and Stochastic Gradient Descent Classifier, were used to predict the intervention that the participant was in based on each trait by

itself, a combination of each trait, PCAs for the traits, and the total change in all outcomes, as well as the PCAs for the changes in outcomes. A leave one out cross validation (LOOCV) was used for model validation [65]. Due to the LOOCV method, $R^2$ cannot be reported in the regressor models, and due to several changes in outcome scores being 0, Mean Absolute Error (MAE) was used to assess model predictive accuracy [66,67]. Classifier models were assessed for accuracy, precision, and recall. Due to small sample sizes and several of the models having 0% accuracy, only accuracy scores are reported, as precision, recall, Area Under the Curve Receiving Operating Characteristics (AUC ROC), and F1 scores were unable to be calculated [68].

## 3. Results

All results can be found in Supplementary Table S1. Only significant results are presented below (Table 2). All descriptive data are presented in Supplementary Tables S2–S5.

**Table 2.** Significant Findings Only.

| Factor | Measure | F-Stat | *p*-Value | $\eta^2_G$ |
|---|---|---|---|---|
| Trait Physical Energy × Intervention × Time | Concentration Task Grid Score | 2.779 | 0.041 | 0.030 |
| Time | Distractors correct | 3.354 | 0.050 | 0.016 |
| Time | Distractors average reaction time | 5.215 | 0.014 | 0.010 |
| Trait Physical Energy × Intervention | Forward Counting Correct | 6.730 | 0.005 | 0.064 |
| Time | Forward Counting Correct | 3.559 | 0.040 | 0.037 |
| Trait Physical Energy × Intervention | RVIP primary reaction time | 4.234 | 0.029 | 0.036 |
| Time | RVIP primary reaction time | 5.523 | 0.013 | 0.029 |
| Trait Physical Energy × Intervention × Time | Serial addition average reaction time | 3.562 | 0.029 | 0.017 |
| Time | Stroop congruent average reaction time | 25.336 | <0.001 | 0.076 |
| Time | Stroop incongruent average reaction time | 9.895 | 0.002 | 0.057 |
| Trait Physical Energy × Time | Tension | 5.737 | 0.021 | 0.021 |
| Trait Physical Energy × Intervention | Fatigue | 3.860 | 0.031 | 0.088 |
| Intervention × Time | Confusion | 3.770 | 0.020 | 0.043 |
| Time | Vigor | 5.990 | 0.006 | 0.017 |
| Time | State Physical Energy | 13.961 | <0.001 | 0.082 |
| Trait Physical Energy × Time | State Physical Energy | 3.902 | 0.048 | 0.024 |
| Trait Physical Energy × Intervention | State Physical Fatigue | 4.050 | 0.026 | 0.104 |
| Time | State Physical Fatigue | 16.368 | <0.001 | 0.042 |
| Trait Physical Energy × Intervention | State Mental Energy | 4.293 | 0.025 | 0.088 |
| Time | State Mental Energy | 4.85 | 0.018 | 0.011 |
| Time | State Mental Fatigue | 15.119 | <0.001 | 0.056 |
| Trait Physical Energy × Intervention | Motivation | 5.700 | 0.011 | 0.129 |
| Time | Motivation | 10.901 | 0.001 | 0.026 |
| Trait Physical Fatigue × Intervention | Forward Counting Correct | 6.123 | 0.007 | 0.062 |
| Trait Physical Fatigue | RVIP incorrect secondary | 5.370 | 0.031 | 0.084 |
| Trait Physical Fatigue | Stroop percentage correct congruent | 6.746 | 0.016 | 0.036 |
| Trait Physical Fatigue | Stroop correct incongruent | 5.156 | 0.033 | 0.023 |
| Trait Physical Fatigue | Stroop percentage correct incongruent | 5.156 | 0.033 | 0.023 |

**Table 2.** *Cont.*

| Factor | Measure | F-Stat | *p*-Value | $\eta^2{}_G$ |
|---|---|---|---|---|
| Trait Physical Fatigue × Intervention | Depression | 3.618 | 0.045 | 0.102 |
| Intervention × Time | Vigor | 2.658 | 0.048 | 0.011 |
| Trait Mental Energy × Intervention | Forward counting average reaction time | 3.79 | 0.040 | 0.043 |
| Trait Mental Energy | RVIP correct secondary | 4.423 | 0.048 | 0.128 |
| Trait Mental Energy | RVIP incorrect secondary | 4.809 | 0.040 | 0.074 |
| Trait Mental Energy × Intervention × Time | RVIP tertiary percent correct | 3.436 | 0.023 | 0.023 |
| Trait Mental Energy × Intervention | Serial subtract three correct | 3.911 | 0.029 | 0.040 |
| Trait Mental Energy × Time | Serial subtract three attempts | 4.382 | 0.030 | 0.019 |
| Trait Mental Energy | Vigor | 7.095 | 0.015 | 0.101 |
| Trait Mental Energy | State Physical Fatigue | 6.313 | 0.021 | 0.061 |
| Trait Mental Fatigue × Intervention | Distractors correct | 4.160 | 0.025 | 0.036 |
| Trait Mental Fatigue × Intervention | RVIP correct primary | 4.357 | 0.027 | 0.028 |
| Time | RVIP correct tertiary | 3.757 | 0.034 | 0.013 |
| Trait Mental Fatigue × Intervention | RVIP primary omitted | 4.373 | 0.027 | 0.028 |
| Trait Mental Fatigue × Intervention | RVIP primary percent correct | 4.377 | 0.027 | 0.028 |
| Time | RVIP tertiary percent correct | 3.782 | 0.033 | 0.013 |
| Trait Mental Fatigue × Intervention × Time | Serial subtract seven correct | 2.825 | 0.045 | 0.02 |
| Trait Mental Fatigue | Stroop percentage correct incongruent | 4.664 | 0.041 | 0.022 |
| Trait Mental Fatigue × Time | Stroop percentage correct incongruent | 5.917 | 0.019 | 0.046 |
| Time | Fatigue | 6.548 | 0.014 | 0.023 |
| Trait Mental Fatigue × Time | Fatigue | 6.924 | 0.012 | 0.024 |
| Trait Mental Fatigue × Intervention × Time | Fatigue | 3.510 | 0.027 | 0.017 |

Time: For all interventions, participants had significant ($p < 0.05$) declines in the number of correct responses on the distractor tasks (Δ0.58) and the percent of correct responses on the RVIP tertiary task (Δ4.94). All participants also had significantly slower reaction times (Δ0.53 s), RVIP primary task (Δ0.04 s), and on both the congruent (Δ0.07 s) and incongruent (Δ0.11 s) aspects of the Stroop task. All participants also reported significant decreases in feelings of vigor (Δ0.95), decreases in state physical energy (Δ12.94), but increases in state mental energy (Δ12.97) and the motivation to perform mental tasks (Δ23.05). Significant increases in feelings of fatigue (Δ0.92) were also reported; however, when measuring fatigue using the state/trait mental and physical energy and fatigue questionnaire, there were significant improvements in state physical (Δ27.72) and mental (Δ27.36) fatigue over the course of the protocols.

Intervention × Time only: Results suggest that there were non-significant small increases in feelings of confusion over the course of the 90 min in the sitting (Δ0.12) and walking (Δ0.05) interventions; however, individuals had significant declines in confusion in the standing (Δ0.37) intervention. Additionally, while there was only a small decline in feelings of vigor in both the walking (Δ0.54) and sitting conditions (Δ0.64), there was a larger, but significantly greater decline in feelings of vigor over time in the standing condition (Δ1.65).

The present study also finds a significant trait x intervention interaction (F (1.63, 47.17) = 4.234, $p = 0.029$, $\eta^2{}_G = 0.036$) and a significant main effect for time (F (1.47, 42.62) =



5.523, $p = 0.013$, $\eta^2_G = 0.029$) for reaction time on the RVIP primary task. Post hoc analyses suggest that reaction times decreased incrementally over time ($\Delta 0.04$ s) with low trait physical energy individuals having the best reaction time during the walking condition ($0.65 \pm 0.08$ s), while there were no significant differences in reaction time for any other conditions. A significant trait x intervention x time interaction for reaction time on the serial addition task was also found (F (2.91, 84.33) = 3.562, $p = 0.029$, $\eta^2_G = 0.017$) and post-hoc analyses suggest that individuals who report low trait physical energy significantly improved their performance during the walking condition ($-\Delta 0.90$ s), while individuals with high trait energy performed worse ($+\Delta 0.44$ s) in the walking condition. There were no differences in reaction time between groups for the sitting or standing interventions.

Individuals who exhibit high trait physical energy reported significant declines in feelings of anxiety over time ($-\Delta 0.82$), while individuals who were low trait physical energy reported significant increases in feelings of anxiety ($+\Delta 0.16$) over the course of the 1.5 h intervention (F (1.47, 42.62) = 5.737, $p = 0.021$, $\eta^2_G = 0.021$) (Figure 3B). Individuals who are low trait physical energy reported the lowest fatigue (F (1.47, 42.62) = 3.860, $p = 0.031$, $\eta^2_G = 0.088$), during the sitting ($3.97 \pm 3.82$) and standing interventions ($3.80 \pm 3.10$), while individuals who were high trait physical energy reported the highest fatigue in the standing condition ($5.86 \pm 3.43$).

Trait Physical Fatigue: Significant main interactions for trait physical fatigue and intervention (F (1.47, 42.62) = 6.123, $p = 0.007$, $\eta^2_G = 0.062$) were found on the forward counting correct task with low trait individuals performing the best in the standing intervention ($14.04 \pm 4.76$) and the worst when seated ($11.74 \pm 4.72$). Individuals who are high trait physical fatigue also made more errors of commission on the secondary task on the RVIP (F (1, 29) = 5.370, $p = 0.031$, $\eta2G = 0.084$), while individuals who were low trait physical fatigue were more likely to perform worse on both the congruent (F (1, 29) = 6.746, $p = 0.016$, $\eta2G = 0.036$) and incongruent (F (1, 29) = 5.156, $p = 0.033$, $\eta2G = 0.023$) aspects of the Stroop task. Individuals who were high trait physical fatigue reported the lowest feelings of depression during the standing intervention ($0.21 \pm 0.65$), but the highest feelings of depression during the walking intervention ($2.07 \pm 1.11$), while individuals who were low trait physical fatigue reported the lowest feelings of depression during the standing condition ($0.40 \pm 1.06$), and the highest feelings of depression during the sitting condition ($1.25 \pm 1.73$).

Trait Mental Energy: Results suggest that individuals who are low trait mental energy had the slowest reaction times on the forward counting task ($6.43 \pm 2.82$) when they were standing; however, their answers did not significantly differ from individuals who were high trait mental energy in the sitting or walking interventions. Individuals who were high trait mental energy reported significantly higher feelings of vigor ($4.03 \pm 3.14$) than individuals who were low trait mental energy ($2.20 \pm 1.91$). However, individuals who were high trait mental energy also reported significantly higher feelings of physical fatigue ($151.01 \pm 60.21$) compared to individuals who were low trait mental energy ($119.12 \pm 52.06$). Surprisingly, individuals who were low trait mental energy committed less errors of omission ($1.52 \pm 1.99$ vs. $2.54 \pm 2.48$) and had significantly more correct responses ($12.92 \pm 3.30$ vs. $9.85 \pm 4.69$) on the RVIP secondary task (Figure 3C) compared to individuals who were high trait mental energy. Individuals who were low trait energy also had significantly more attempts on the serial subtraction three task ($\Delta 3.12$) over time, while individuals who were high trait energy had significant declines in attempts on the serial subtraction three task ($\Delta 1.92$) over time. Interestingly, individuals who were low trait mental energy did not have a significant ($p = 0.256$) improvement in accuracy of responses nor did individuals who were high trait mental energy have a decline in accuracy of responses. Individuals who were low trait mental energy had the most correct responses on the serial subtraction three tasks ($32.06 \pm 12.27$) in the standing condition, while individuals who were high trait mental energy had the least number of correct responses in the standing conditions ($26.29 \pm 13.34$). While individuals who were high trait mental energy had significant ($p < 0.05$) declines in correct responses on the RVIP tertiary task during all

interventions, individuals who were low trait mental energy only had significant declines in performance in the walking condition, but they did not have any changes in correct responses on the RVIP tertiary tasks during the sitting and standing interventions.

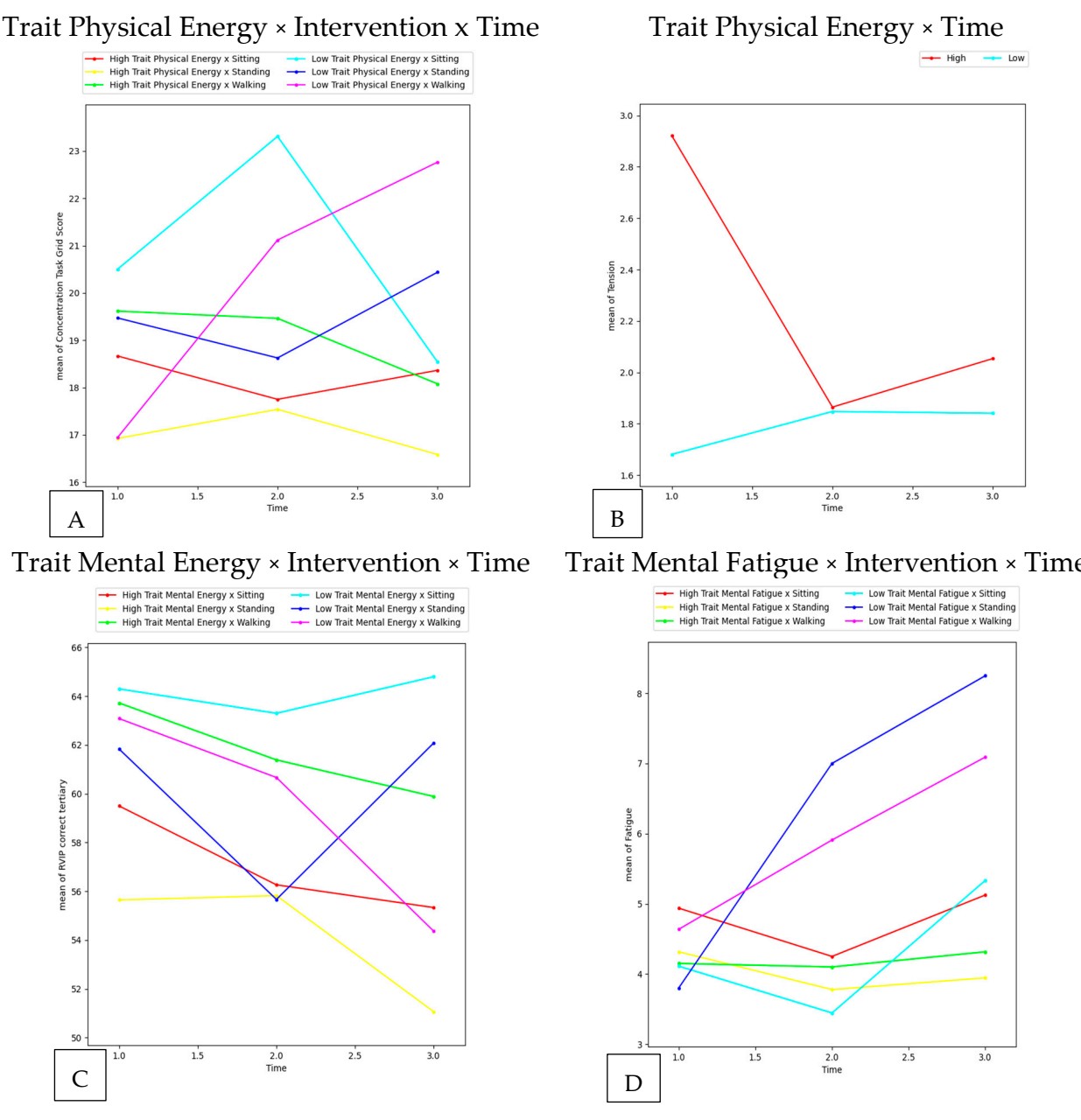

**Figure 3.** (**A**) Trait Physical Energy x Concentration Task Grid $\times$ Time interaction; (**B**) Trait Physical Energy $\times$ Time interaction for POMS Tension; (**C**) Trait Mental Energy $\times$ RVIP correct tertiary task $\times$ Time interaction; (**D**) Trait Mental Fatigue $\times$ POMS Fatigue $\times$ Time interaction.

Trait Mental Fatigue: Individuals who were low trait mental fatigue had significantly more correct responses on the distractors task in the standing intervention ($23.54 \pm 10.63$); however, their scores in the other two interventions did not differ from the high trait mental fatigue individuals. Individuals who were high trait mental fatigue had the more correct responses ($11.53 \pm 3.88$) and lower errors of omission ($4.47 \pm 3.88$) on the RVIP primary task in the sitting intervention compared to individuals who were low trait mental fatigue ($8.52 \pm 4.72$, $7.48 \pm 4.73$), while there were no significant differences between groups in the walking or standing interventions. Individuals who were low trait mental

fatigue performed significantly worse during their first attempt (91.78% $\pm$ 22.23) on the incongruent Stroop task; however, after the initial attempt, their scores did not significantly differ from individuals who are high trait mental fatigue. While there was a significant increase in feelings of fatigue ($\Delta$0.92) over the course of the protocol, individuals who were low trait mental fatigue had a significant increase in feelings of fatigue ($\Delta$2.66), while individuals who are high trait mental fatigue did not experience any changes in feelings of fatigue ($\Delta$0.003). Interestingly, individuals who were low trait mental fatigue reported significant increases in feelings of fatigue during the standing ($\Delta$4.45) and walking interventions ($\Delta$2.45), and they experienced a small, but not significant increase in feelings of fatigue during the sitting intervention ($\Delta$1.22). Individuals who were high trait mental fatigue reported no significant changes in feelings of fatigue during any of the interventions (Figure 3D).

Post hoc machine learning: In the PCA analyses for traits, the traits were reduced to two PCs (PC$_1$ = 1.71 (explained variance = 42.62%), PC$_2$ = 1.25 (explained variance = 31.23%)). The loading for PC$_1$ was Trait Physical Energy = 0.47, Trait Physical Fatigue = $-0.50$, Trait Mental Energy = 0.44, Trait Mental Fatigue = $-0.58$. The loading for PC$_2$ was Trait Physical Energy = $-0.44$, Trait Physical Fatigue = $-0.56$, Trait Mental Energy = $-0.62$, Trait Mental Fatigue = $-0.33$. A total of 52 missing data were imputed for the machine learning analysis. All results for the regressor and classifiers can be found in the Supplementary Materials section. The MAE for the regressors improved when trait moods were added to the models, with MAEs for the various outcomes different based on the trait or combination of traits added to the model. For example, when comparing the sitting and standing conditions, the MAE for the Erikson Flanker was 0.316; however, when traits physical energy and physical fatigue were added to the model, the MAE improved to 0.308. However, state physical fatigue had a MAE of 0.771 and when trait physical energy was added to the model, the MAE value improved to 0.684. When assessing the classifier predictive accuracy, we find that the results are also trait-dependent, as some traits improve the models (i.e., Trait Physical Energy increases predictive accuracy of using POMs Tension to predict the standing vs walking condition from 67.74% to 69.35%), while others reduce predictive accuracy (i.e., Trait Physical Fatigue reduces the predictive accuracy of using POMS Tension scores to predict the standing vs walking condition from 67.74% to 61.29%).

## 4. Discussion

To our knowledge, this is the first study to identify inter-individual responses to the performance of mental tasks using a sitting and standing desk, as well as when sitting is interrupted by intermittent walking. Further, this study is the first to use machine learning to predict changes in various cognitive and mood outcomes using trait mental and physical energy and fatigue. The current findings add significantly to the literature, as they may help explain some of the inconsistent results that have previously been reported for both standing desks [14] and studies using similar mental task protocols to understand various nutritional interventions [10,11,24,25,45,46,69]. Our findings support the use of the low-cost and time efficient survey to measure trait mental and physical energy and fatigue in identifying inter-individual responses to various interventions.

The need to measure trait mental and physical energy and fatigue is best illustrated when evaluating the interesting results for changes in POMS Fatigue. Our study finds that regardless of intervention, there is a small but significant increase in feelings of fatigue over the course of the 1.5 h of the performance of mental tasks. However, when the data are examined more closely, individuals who are low trait mental fatigue reported a significant increase in feelings of fatigue with the performance of mental tasks, and when examining the data between interventions, the biggest increase in feelings of fatigue for low trait mental fatigue individuals occur during the standing and walking interventions, with small but not significant increases in fatigue in the sitting condition. Interestingly, there were no significant differences in feelings of fatigue between interventions. This suggests that if we had not differentiated participants by trait moods, we might have incorrectly stated

that performing cognitive tasks increases feelings of fatigue, but there are no significant differences between interventions. By splitting the groups based on their traits, we were able to identify that low-trait mental fatigue individuals report significant increases in feelings of fatigue in the standing and walking interventions, but individuals who are high trait mental fatigue report no changes in feelings of fatigue. The null findings by Huseman and colleagues [70], who report a change in moods for their participants, but report no significant differences between sitting or standing desk interventions, may warrant further consideration. Specifically, perhaps the findings of Huseman and colleagues [70] might have changed if their methodology had accounted for trait mental and physical fatigue.

The current results are in line with previous studies reporting an increase in POMS confusion when performing cognitive tasks while seated [11] and significant improvement in POMS confusion scores when performing cognitive tasks while standing [71]. However, the findings of Pronk and colleagues [71] were based on the chronic use of standing desks across several weeks, while the present findings suggest that standing desk use may reduce feelings of confusion acutely. Our findings suggest that use of standing desks in a classroom setting may provide an easy to implement method of reducing feelings of confusion felt by participants performing 1.5 h of cognitive tasks.

We also report that while there was a decline in feelings of vigor during all 3 interventions, with the standing intervention resulting in the largest decline in feelings of vigor compared to the sitting and walking interventions. Interestingly, when examining mental and physical energy separately, we find that performing mental tasks while standing increase mental energy; however, feelings of physical energy declined. This nuance may help explain the findings of previous work examining standing desks and moods that supports the chronic use of standing desks to increases feelings of vigor [14]. The previous studies did not differentiate between mental and physical energy [14] and may have captured increases in mental energy in their data. However, one should not discount that repeated exposure to standing desks may increase feelings of physical energy over time.

The present findings support the use of this instrument to understand trait-state mental and physical energy and fatigue interactions. For example, we find that individuals who are normally not physically energetic (low trait physical energy) report significantly larger declines in feelings of physical energy when performing this cognitive battery compared to those who reported feeling high trait physical energy. Similar interactions are also noted for trait and state mental fatigue, with low trait mental fatigue (individuals who are normally not mentally fatigued) reporting significantly higher feelings of fatigue after performance of mental tasks, results consistent with findings from a previous study [24]. These findings are in line with previous literature that has studied trait–state relationships for other moods such as anxiety [72], depression [73], and anger [74]. The current findings also support the mental energy framework proposed by O'Connor [75], as individuals who are low trait energy report significantly lower feelings of mental energy and motivation at baseline, and over the course of the 1.5 h protocol, the increases in mental energy and motivation are highly correlated (R = 0.90). Interestingly, similar to results from a previous study [24], scores on the POMS Fatigue scale were opposite to the state physical and mental fatigue measurements in the current study. With high $\alpha$'s in both investigations ($\alpha > 0.80$) on both the POMS and the State Mental and Physical Energy and Fatigue surveys, the results are consistent yet quizzical.

Many of the findings from the current study suggest that individuals with low trait mental or physical energy improve in their cognitive task performance over the course of the protocol, a finding consistent with the mental energy framework [75]. However, the nuances in the changes suggest that the sitting and standing interventions provide the most benefit in improving accuracy of response, while walking and standing improve speed of response. Together, these findings contradict the findings by Huseman and colleagues [70], who report a non-significant decline in efficiency of task performance in the standing condition. Our findings suggest that this decline in efficiency may be primarily driven

by individuals who are high trait energy, while individuals who are low trait energy may greatly benefit from performing cognitive tasks while standing.

As expected, individuals who were low trait mental energy performed significantly worse than high trait mental energy individuals on the serial subtraction 3 times at baseline, a finding consistent with a previous study examining the use of caffeine [25]. While the previous study measured the serial subtraction 3 responses to caffeine [25], in the present study, high trait mental energy individuals had a significant decline in the number of attempts over time, and low trait mental energy individuals experienced the opposite effect on their performance. Similar changes are also noted on other cognitive tasks and moods, with many of those outcomes being significant prior to the FDR correction of our analyses. Taken together, these findings suggest that individuals who do not normally feel energetic (both mental and physical) may benefit from performing cognitive tasks, especially when using a standing desk.

Although not appropriately powered, the exploratory post hoc analyses of our study find some very interesting results. These findings suggest that although individuals are a combination of all four trait level moods, using individual traits may be better at predicting individual responses to the sitting, standing, and walking conditions. When predicting overall cognitive responses (PCA of outcomes), a combination of trait moods, specifically, trait mental and physical fatigue may be the best for predicting cognitive and mood outcomes to these interventions. The classifier models also had similar results, in that if using individual outcomes, individual traits were better at predicting the intervention the individual was in, while using the PCA of the outcomes, the PCA of the traits improved predictive accuracy of predicting which intervention the individual was participating in. These exploratory results may provide a roadmap for future researchers who may be interested in trying to predict inter-individual responses to various interventions.

As is the case with all studies, the present study is not without limitations. The first limitation was the fact that behaviors from the previous day were self-reported, with previous evidence suggesting that college students in general tend to over-estimate their positive behaviors [76]. Secondly, while the primary analyses were properly powered, the post hoc analyses were not, suggesting that the results of the machine learning analyses should be interpreted with caution. Further, there were 52 missing pieces of data in this study, which may have impacted the results of some of the mixed ANOVA analyses. However, considering these data accounted for 0.32% of the total data collected in this study, the results may still be properly interpreted. Another potential limitation of generalizability of this study is that participants were primarily female. Future studies should try to have a more balanced sample of men and women. In addition, while the use of an assessment tool such as the National Institute of Health (NIH) cognitive toolbox [77] may have served as an appropriate method for creating cognitive profiles, the trait mental and physical energy and fatigue survey provided a no-cost alternative that could be completed in approximately 30 s, which is significantly faster than the 30 min it usually takes to complete the free for use NIH cognitive toolbox. Finally, another potential limitation of this study is that participants were not asked whether they used standing desks and/or took short walking breaks when they were studying or doing cognitively demanding work in their daily life. Individuals who may have been regularly using standing desks may have adjusted to the use of standing desks during the provided breaks and reported either no significant changes or improvements in moods and cognitive responses [71].

*Practical Implications*

While this is the first study to try to identify acute inter-individual responses to sitting desks, standing desks, and intermittent bouts of 2 min of walking among college students, these findings may provide some implications for both students and office workers. Individuals interested in modifying their cognitive task performance or moods over the course of a school or workday may want to consider their usual feelings of energy and fatigue (trait level moods) when deciding on an intervention. For example, individuals

who may not normally feel energetic may want to explore the use of standing desks when completing their work, as this study provides evidence that these individuals may benefit from the use of a standing desk. Further, individuals who may have low trait mental fatigue may benefit from performing mental tasks in a seated position, as it would be the least fatiguing of the conditions tested. Finally, another potential implication of our findings is that individuals who normally report being highly energetic may benefit from performing mental tasks, as it may reduce their feelings of anxiety. However, as this is just a single study, additional studies need to be conducted to continue to further investigate these findings.

### 5. Conclusions

This study identified inter-individual responses in college students during performance of cognitive tasks when performed on a sitting desk, standing desk, and when interrupted by 2 min of walking. The findings suggest that trait mental and physical energy and fatigue may explain the inter-individual differences in cognitive task performance and mood responses in the three interventions. Further, the post hoc machine learning analyses support the use of trait mental and physical energy and fatigue to predict changes in cognitive task performance and moods in sitting desks, standing desks, and when sitting is interrupted by intermitted walking. Future researchers should try to measure trait mental and physical energy and fatigue prior to administering other interventions to determine whether these traits may be able to predict inter-individual responses to other interventions as well.

**Supplementary Materials:** The following supporting information can be downloaded at https://www.mdpi.com/article/10.3390/app13074241/s1, Table S1: All Results, Table S2: Trait Physical Energy, Table S3: Trait Physical Fatigue, Table S4: Trait Mental Energy, Table S5: Trait Mental Fatigue.

**Author Contributions:** Conceptualization: A.B., C.H., M.R. and B.M.; methodology: A.B., C.H., M.R. and B.M; validation: A.B. and H.M.G., formal analysis: H.M.G. and A.B.; investigation: C.H., M.R. and B.M., data curation: C.H., M.R. and B.M.; writing—original draft preparation: H.M.G., C.L.W.-R., J.M., C.H., M.R. and B.M.; writing—reviewing and editing: H.M.G., C.L.W.-R., J.M., C.H., M.R., B.M. and A.B.; visualization: H.M.G. and A.B.; supervision: A.B.; project administration: A.B. All authors have read and agreed to the published version of the manuscript.

**Funding:** This research received no external funding.

**Institutional Review Board Statement:** The study was conducted in accordance with the Declaration of Helsinki and approved by the Institutional Review Board (or Ethics Committee) or Clarkson University (#18–49.2, 16 May 2020).

**Informed Consent Statement:** Informed consent was obtained from all subjects involved in the study.

**Data Availability Statement:** The data presented in this study are available on request from the corresponding author. The data are not publicly available due to IRB restrictions.

**Conflicts of Interest:** The authors declare no conflict of interest.

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
