# Peer review of "Trait Energy and Fatigue Influence Inter-Individual Mood and Neurocognitive Responses during Work Done While Sitting, Standing, and Intermittent Walking: A Randomized-Controlled Crossover Design"

_applsci, doi:10.3390/app13074241_

Round 1

Reviewer 1 Report

In this study the authors examine t the inter-individual cognitive performance and mood changes of college students during the performance of a cognitive task battery, while seated, standing and with intermittent bouts of walking.

Although the study has the potentiality of being shared with the scientific community, I believe that the manuscript would benefit from a minor revision with the attempt to better support their experimental setting.

 1. Abstract should start with a paragraph dedicated to a brief description of  the background.

2. More information should be provided about the Experimental

3. The intervention protocol should be better described.

4. More information should be provided about the participants’ characteristics.

5. The Discussion should be enriched with the existing theory. The authors should clearly describe the scientific evidence that supports their findings.

6. I would like to see more of the practical implications. Based on the analyzed variables, how the authors intend to use their findings? 

Kind regards

Author Response

We are pleased to re-submit the following manuscript titled: “Trait Energy and Fatigue influence inter-individual mood and neurocognitive responses during work done while sitting, standing, and intermittent walking: A randomized-controlled crossover design”. We believe we have fully addressed the revisions requested. Additionally, we would like to express our appreciation to the reviewer for providing the feedback. Below are the reviewer comments and our responses in red.

In this study the authors examine the inter-individual cognitive performance and mood changes of college students during the performance of a cognitive task battery, while seated, standing and with intermittent bouts of walking.

Although the study has the potentiality of being shared with the scientific community, I believe that the manuscript would benefit from a minor revision with the attempt to better support their experimental setting.

  1. Abstract should start with a paragraph dedicated to a brief description of  the background.

Response: We have now added a short sentence to provide a brief description of the background.

  1. More information should be provided about the Experimental

Response: We appreciate the reviewer’s comment, but in light of comments made by another reviewer that we provided too much information when describing our experiment, providing insight into what specific information they feel is missing may help us to address this concern more appropriately.

  1. The intervention protocol should be better described.

Response: Again this feedback is appreciatedbut as is the case with the previous comment, we would appreciate guidance from the reviewer about the additional information they believe would strengthen the manuscript. 

  1. More information should be provided about the participants’ characteristics.

Response: In response to this comment we have now  provided additional information about participant characteristics in Table 1. If the reviewer believes it to be appropriate we could also describe additional  participant characteristics in our methodology section, however, we did not want to duplicate the information. As stated above, another reviewer has suggested cutting down on our methodology. so we would appreciate it if the reviewer could clarify what other information we could provide.

  1. The Discussion should be enriched with the existing theory. The authors should clearly describe the scientific evidence that supports their findings.

Response: We appreciate the reviewer’s feedback and have consistently tied our findings back to previous literature. While the existing theory on trait mental and physical energy and fatigue is still new, most of the work that has been conducted on how trait influences state has been done in the field of anxiety, depression and anger. Therefore, we have added additional references related to how trait anxiety, depression and anger influence the state aspects of those moods, which is in line with our findings on trait-state energy and fatigue. Further, we have referenced the work of P. O’Connor on the trait-state energy and fatigue relationship. In the introduction, we adress potential biological and neurological mechanisms that are uniquely associated with trait-state energy and fatigue to further provide evidence of why we explored trait mental and physical energy and fatigue. We did not think that speculating on the potential biological mechanisms that may have influenced our findings was appropriate in our discussion section. However, we do understand the reviewer’s point, which is why we have provided the evidence on the biological and potential neurological correlates of the traits in the introduction and also supported our findings by tying our current findings back to the published work fro researchers who have studied trait-state interactions in anxiety, depression and anger. 

  1. I would like to see more of the practical implications. Based on the analyzed variables, how the authors intend to use their findings? 

Response: We appreciate the reviewer’s comments and have now added an implications section to our manuscript as part of the discussion section. 

Reviewer 2 Report

This is an interesting although not compelling paper. The authors present their study in great detail, perhaps almost too much detail.  It should be possible to summarize information about each test used and make additional information available to interested readers in a supplement or to put it in tabular form. There are also several ways to create cognitive profiles, standard scores to facilitate comparisons and/or further categorize the test results. There are a few design concerns, particularly the gender imbalance (the sample is primarily female) and the small sample which precludes the conduct of some analyses.  Additionally, the definition of "mood" is focused upon energy and fatigue, usually used as indicators of mood rather than mood itself.  Since this is a core concept in this study, further discussion of energy and fatigue would provide a stronger theoretical foundation for this study.  Finally, the usefulness (future application) of the results is limited by the choice of participants.  The results of this study would be of far more interest to office workers of many types who spend the greater part of their day sitting at a desk.

Author Response

This is an interesting although not compelling paper. The authors present their study in great detail, perhaps almost too much detail.  It should be possible to summarize information about each test used and make additional information available to interested readers in a supplement or to put it in tabular form.

Response: We appreciate the reviewer’s feedback regarding our study and while agree with the reviewer that the methods are presented in excruciating detail, we have had another reviewer suggest that we did not provide enough detail. Therefore, we are happy to work with the editor to determine what information is appropriate to keep in manuscript, what information should be augmented, and what information should move to a supplementary table.

There are also several ways to create cognitive profiles, standard scores to facilitate comparisons and/or further categorize the test results.

Response: We acknowledge this reviewer’s concerns that there are several ways to create cognitive profiles, especially with tools such as the NIH Toolbox. However, we were interested in trying to create profiles of individuals that did not require ~30 minutes of cognitive testing and were not cost prohibitive (i.e. free).This approach allowed our team to determine whether a no-cost alternative, such as the trait level mental and physical energy and fatigue questionnaire, would allow us to create profiles that could predict cognitive task and mood responses. We have now clarified this in our introduction and discussion sections.

There are a few design concerns, particularly the gender imbalance (the sample is primarily female) and the small sample which precludes the conduct of some analyses. 

Response: We agree with the reviewer’s concerns about the gender imbalance and have now addressed this in our limitations section. We also agree that the sample size was small for the machine learning models, which is why we have stated that we used the machine learning analyses as an exploratory method to see if the addition of the trait mental and physical energy and fatigue scales would help predict the changes in responses. As it relates to our primary analysis, the study was properly powered and our post-hoc power analyses were >0.8 for all significant findings.

Additionally, the definition of "mood" is focused upon energy and fatigue, usually used as indicators of mood rather than mood itself. 

Response: While our study did use trait level mental and physical energy and fatigue moods, our state moods included anxiety, depression, anger, confusion and energy and fatigue. Energy and fatigue state moods are classified as mood states by  a number of researchers, including W. Morgan, Dishman, O’Connor, McNair, Lorr, Raglin and many other well published scholars in the field of psychology. Additioally, energy and fatigue have been classified as moods, and not only as indicators of mood, by multiple scales that measure mood states, including the POMS, AD-ACL, SF-36 and the O'Connor Mental and Physical Energy and Fatigue State-Trait scale, that was used in this study. Below are some well cited peer review papers that provide evidence of energy and fatigue as mood states.

O'Connor, P. J. (2004). Evaluation of four highly cited energy and fatigue mood measures. Journal of psychosomatic research57(5), 435-441.

McNair, D. M., & Lorr, M. (1964). An analysis of mood in neurotics. The Journal of Abnormal and Social Psychology69(6), 620.

Lorr, M., & Wunderlich, R. A. (1988). A semantic differential mood scale. Journal of clinical psychology44(1), 33-36.

O'Connor, P. J. (2006). Mental energy: Assessing the mood dimension. Nutrition reviews64(suppl_3), S7-S9.

Kenttä, G., Hassmén, P., & Raglin, J. S. (2006). Mood state monitoring of training and recovery in elite kayakers. European Journal of Sport Science6(4), 245-253.

Morgan, W. P. (1980). The trait psychology controversy. Research Quarterly for Exercise and Sport51(1), 50-76.

Since this is a core concept in this study, further discussion of energy and fatigue would provide a stronger theoretical foundation for this study. 

Response: We appreciate the reviewer’s feedback and have now provided significant evidence on the biological and biomechanical differences in energy and fatigue as part of our introduction. However, we can only speculate as to how the biological differences in these moods may influence responses to the interventions in our study. Therefore, we chose not to speculate on the potential biological reasons for how trait level mental and physical energy and fatigue may have influenced the outcomes in our study.

Finally, the usefulness (future application) of the results is limited by the choice of participants.  The results of this study would be of far more interest to office workers of many types who spend the greater part of their day sitting at a desk.

Response: While we agree that these results have implications beyond university classrooms and may be of interest to those whose occupation requires daily seated work, we initiated this study because the students in the health sciences programs at our institutions spend approximately 8 to 10 hours a day in classes during the didactic portion of their studies and we believe this has implications on their mood state and cognitive function. Previous studies by our lab suggest that the sedentary aspects of their studies were negatively influencing moods and other health biomarkers. Therefore, we were motivated to explore interventions that our students could easily adopt in the classroom and have framed our discussion to this end.

Author Response

The Authors explored inter-individual responses in a group of college students while performing a combination of cognitive tasks with standing, sitting, and intermittent walking. Emphasis is given to trait mental and physical energyand fatigue questionnaires as predictors of inter-individual responses to performed tasks. Results supported the use ofstanding desks for better cognitive performances.

I feel that the topic of this study is relevant and of general interest, I only have few minorcomments detailed below.

I would suggest that the manuscript should be formatted with line numbers. A more feasible review could be done with the reported line/s numbers.

Response: We have now added line numbers.  Thank you for this suggestion.

Minor comments:

Abstract

The abstract should be structured, but without headings. Please, revise it as a single paragraph by removing the words “Objective”, “Methods”, “Results”, and “Conclusion”.

Response: The headings have been removed from the abstract.

Methods

The underlining of headings in the methods section are different (e.g., spaces between words and colon). Please, revisethem by choosing one and uniform all of them throughout the manuscript.

Response: We have addressed this and underlining and headings are uniform throughout the manuscript.

Study design

There is an additional space at the beginning of the paragraph that should be removed to uniform with the other paragraphs. The same issue at the end of paragraph between “[…] mental task battery trials.” and “Participants completed…”.

Response: These space issues have been fixed. Thank you for bringing out attention to this issue.

Participants

There is an additional space at the beginning of the paragraph (before the word “Approval….”) and in the middle (before “of the 40 volunteers…”) that should be removed.

Response: These space issues have been fixed.

“in case there were outliers and data had to be excluded. No data were excluded from this study…”. Consider adding “Accordingly, no data were excluded…”.

Response: Thank you for the suggestion. The edit has been made.

Table 1: data in the table 1 should be formatted with the same font of the main text. Also “hours per week” or “hours/week” should be revised as the Authors prefer, and round brackets closed after the word “week” of both vigorous and moderate physical activity.

Response: We appreciate the reviewer’s attention to detail and as such these edits have been made.     

There is an additional space after the Table 1 (before “The average self-reported…”) that should be removed.

Response: The space issue has been fixed.

and 23 of the 31 participants had PSQI scores > 5.”: please, explain what PSQI stands for in the text before using the abbreviation.

Response: PSQI is  fully described as the Pittsburg Sleep Quality Index in its first use.

Testing Day Measures

Consider removing the space between the paragraphs “Pre-testing measures” and “Mental Task Battery”.

Response: The edit has been made.

Consider using only one between the round brackets or the full stop in the bulleted list numbers.

Response: The edit has been made to remove the full stop. Thank you.

The number 4) “Distractors” has a different hyphen that should be uniformed with the others in the bulleted list numbers.

Response: The edit has been made.

Mood and Motivation Surveys

Consider removing the colon after the word “surveys” and the space between the title and the first bulleted number.

Response: The edit has been made.

Procedure

Figure 2: I would ask if the reported time of tasks is a mean of completion time among participants or exactly the time required to complete each task, or other. It could be useful to add this information in the figure legend.

Response: The reported time of tasks are mean completion times. The rest times were fixed for all participants. We’ve added a legend to the figure to clarify.  Thank you for this suggestion.

Results

Table 2: data in the table 2 should be formatted with the same font of the main text.

Response: The edit has been made.

The underlining of headings in the results section are different (e.g., spaces between words and colon). Please, revisethem by choosing one and uniform all of them throughout the manuscript.

Response: ADD

“walking (Δ0.05) interventions,/ however, individuals had significant declines in confusion”: the slash should be removed.

Response: The edit has been made.

Figure 3: I think that Figure 3 is a screenshot of a word file and I would suggest to change it with a jpg format, with better resolution; legend and axes titles cannot be read well.

Response: We have now put the .png figures that were created by matplotlib into the document.

Trait Physical Energy paragraph: please, remove the additional spaces at the beginning of the periods “The present study also finds a significant trait x intervention interaction…” and “Individuals who exhibit high trait physical energy reported significant…”

Response: We have removed the additional spaces.

Discussion

Please, remove the additional spaces at the beginning of and throughout the paragraph.

Response: The edit has been made.

“…posed by O’Connor (61), asindividuals who are low trait energy report significantly…”: it should be added a space to separate “as” and “individuals”.

Response: The edit has been made.

The statements regarding author contributions, funding, etc., should be completed.

Response: We have added these statements.

References

References must be numbered in order of appearance in the text. For example, in the introduction section, it cannot be found the references from 8 to 12.

Response: We have corrected this.

The Reference n. 16 is thought to be matched with Boolani et al., however, in the reference list it is reported a work of Loy BD et al. The same issue is with other references (e.g., n. 25, 26, etc.) Please, check carefully the references in the text and the correct match with the references in the list.

Response: Thank you for this correction.  These corrections have been made.

References numbers in the main text should be placed in square brackets [ ]. Also, they are also incorrectly formatted orwritten in the reference list. Please, revise them according to the requested Journal references’ style (e.g., see below).

Response: We have corrected this. 

Round 2

Reviewer 2 Report

The authors have provided valuable additional information to the manuscript.